# Artificial Intelligence-Based Segmentation of Residual Tumor in Histopathology of Pancreatic Cancer after Neoadjuvant Treatment

**DOI:** 10.3390/cancers13205089

**Published:** 2021-10-12

**Authors:** Boris V. Janssen, Rutger Theijse, Stijn van Roessel, Rik de Ruiter, Antonie Berkel, Joost Huiskens, Olivier R. Busch, Johanna W. Wilmink, Geert Kazemier, Pieter Valkema, Arantza Farina, Joanne Verheij, Onno J. de Boer, Marc G. Besselink

**Affiliations:** 1Department of Surgery, Amsterdam UMC, Cancer Center Amsterdam, University of Amsterdam, 1081 HV Amsterdam, The Netherlands; b.v.janssen@amsterdamumc.nl (B.V.J.); r.t.theijse@amsterdamumc.nl (R.T.); s.vanroessel@amsterdamumc.nl (S.v.R.); o.r.busch@amsterdamumc.nl (O.R.B.); 2Department of Pathology, Amsterdam UMC, Cancer Center Amsterdam, University of Amsterdam, 1081 HV Amsterdam, The Netherlands; p.valkema@amsterdamumc.nl (P.V.); a.farina@amsterdamumc.nl (A.F.); j.verheij@amsterdamumc.nl (J.V.); o.j.deboer@amsterdamumc.nl (O.J.d.B.); 3SAS Institute Besloten Vennootschap, 1272 PC Huizen, The Netherlands; Rik.deRuiter@sas.com (R.d.R.); Antonie.berkel@sas.com (A.B.); Joost.Huiskens@SAS.com (J.H.); 4Department of Medical Oncology, Amsterdam UMC, Cancer Center Amsterdam, University of Amsterdam, 1081 HV Amsterdam, The Netherlands; j.w.wilmink@amsterdamumc.nl; 5Department of Surgery, Amsterdam UMC, Cancer Center Amsterdam, Vrije Universiteit, 1081 HV Amsterdam, The Netherlands; g.kazemier@amsterdamumc.nl

**Keywords:** pancreatic cancer, histopathology, tumor response scoring, neoadjuvant therapy, artificial intelligence, machine learning

## Abstract

**Simple Summary:**

The use of neoadjuvant therapy (NAT) in patients with pancreatic ductal adenocarcinoma (PDAC) is increasing. Objective quantification of the histopathological response to NAT may be used to guide adjuvant treatment and compare the efficacy of neoadjuvant regimens. However, current tumor response scoring (TRS) systems suffer from interobserver variability, originating from subjective definitions, the sometimes challenging histology, and response heterogeneity throughout the tumor bed. This study investigates if artificial intelligence-based segmentation of residual tumor burden in histopathology of PDAC after NAT may offer a more objective and reproducible TRS solution.

**Abstract:**

Background: Histologic examination of resected pancreatic cancer after neoadjuvant therapy (NAT) is used to assess the effect of NAT and may guide the choice for adjuvant treatment. However, evaluating residual tumor burden in pancreatic cancer is challenging given tumor response heterogeneity and challenging histomorphology. Artificial intelligence techniques may offer a more reproducible approach. Methods: From 64 patients, one H&E-stained slide of resected pancreatic cancer after NAT was digitized. Three separate classes were manually outlined in each slide (i.e., tumor, normal ducts, and remaining epithelium). Corresponding segmentation masks and patches were generated and distributed over training, validation, and test sets. Modified U-nets with varying encoders were trained, and F1 scores were obtained to express segmentation accuracy. Results: The highest mean segmentation accuracy was obtained using modified U-nets with a DenseNet161 encoder. Tumor tissue was segmented with a high mean F1 score of 0.86, while the overall multiclass average F1 score was 0.82. Conclusions: This study shows that artificial intelligence-based assessment of residual tumor burden is feasible given the promising obtained F1 scores for tumor segmentation. This model could be developed into a tool for the objective evaluation of the response to NAT and may potentially guide the choice for adjuvant treatment.

## 1. Background

Neoadjuvant therapy (NAT) is increasingly used for patients with locally advanced and (borderline) resectable pancreatic ductal adenocarcinoma (PDAC). Recent clinical studies have shown that NAT affects overall survival, disease-free survival, and margin-negative resection rates positively [1,2,3,4,5,6]. Histologic examination of PDAC resection specimens following NAT offers the opportunity to evaluate treatment response using various tumor response scoring (TRS) systems [7,8]. TRS is believed to serve at least two important purposes [7]. First, it may be used in clinical trials to compare the efficacy of different neoadjuvant regimens in PDAC. Secondly, TRS may guide the choice for adjuvant therapy in the individual patient. It is imperative that TRS accurately correlates with the oncological outcome (i.e., overall survival) for both purposes.

Over the last few decades, researchers proposed several histopathological TRS systems for PDAC to evaluate NAT responses [8,9,10]. Still, during the 2019 Amsterdam international consensus meeting on histological assessment of tumor response of resected pancreatic cancer after NAT, the international study group of pancreatic pathologists (ISGPP) stated that most TRS systems suffer from flawed reasoning or lack objective definition criteria [7]. Most fundamentally, the interobserver agreement is highly insufficient for the most used systems or has not been widely determined, likely related to the subjective nature of the criteria used to define the different categories. Some TRS systems evaluate the ratio of vital tumor rests versus treatment-induced fibrosis. However, distinguishing between treatment-induced fibrosis and desmoplasia or pancreatitis-related fibrosis is at the very least challenging and may, in fact, be impossible with the naked eye. Because of these limitations, the ISGPP stated that a new TRS system should assess residual (viable) tumor burden instead of tumor regression, and objective criteria for the different categories are needed. Artificial intelligence (AI)-based techniques have the potential to fulfill these needs. Artificial intelligence models may be developed to automatically segment and quantify residual tumor burden in histological sections of neoadjuvantly treated and resected PDAC, potentially providing the basis for an objective TRS system.

In this study, we investigated if AI-based segmentation of residual tumor burden in histopathological slides after resection of PDAC after NAT is feasible and may offer a foundation for a more objective and reproducible solution for TRS. To this end, we report developing an AI-based segmentation tool in PDAC segmenting residual tumor burden in histopathological slides of patients following NAT to study treatment response.

## 2. Materials and Methods

### 2.1. Data Acquisition

We retrospectively collected histopathological hematoxylin and eosin (H&E)-stained slides of pancreatic cancer resection specimens of neoadjuvantly treated patients from the archive of the Department of Pathology at Amsterdam UMC in the Netherlands. Any amount or type of neoadjuvant chemo(radio)therapy was considered suitable to be included in this study. Per patient, one representative H&E slide of the tumor bed was selected by A.F. or J.V. and digitized using a Philips Intellisite Ultra-Fast Scanner (Philips, Best, The Netherlands). Whole-slide images (WSI) were converted to BigTiff format and downloaded from the image management system.

### 2.2. Data Handling

First, an expert pathologist (A.F. or J.V.) marked a representative region of interest (ROI) on the WSI using the ‘ASAP’ software package [11]. Within the boundaries of these ROI, histopathological structures were manually outlined at the pixel level using ASAP with coordinates stored as XML. The following structures were annotated: (1) normal ducts; (2) cancerous ducts; (3) in situ neoplasia; (4) islets of Langerhans; (5) acinic tissue; (6) atrophic metaplastic parenchyma; (7) fat; (8) vessels; (9) nerves; and (10) lymphocytic infiltrates. Detailed annotations were prepared in ASAP by B.J., R.T., and A.F., and when made by non-pathologists (B.J. and R.T.), they were evaluated and, if necessary, corrected by expert pancreas pathologists. If a case was considered too difficult to annotate manually reliably, it was excluded from further analysis.

The ROIs were converted from BigTiff to PNG format, and the same area ground-truth annotation-based segmentation masks were generated as PNG files. Images and corresponding masks were generated at a resolution of 0.5 µm/px (‘20×’). The original classifications were pooled into three new classes for further analysis: (1) normal ducts; (2) cancerous ducts and in situ neoplasia combined; and (3) the remaining non-tumorous epithelial tissue (NTET), consisting of islets of Langerhans, acinic tissue, and atrophic metaplastic parenchyma. The remaining classes (fat, vessels, nerves, and lymphocytic infiltrates) were ignored and considered as background elements. Using a sliding window approach, partly (50%) overlapping patches of 512 by 512 pixels were generated from the H&E and corresponding mask image. Patches and corresponding mask images were only included in the dataset if at least 10% of the patch’s surface area was occupied by one of the segmentation classes. The maximum number of patches for each class was limited to 100 per case to limit class imbalance in the training data.

### 2.3. Machine Learning

A ‘standard’ U-net [12] and a selection of modified U-nets with different encoders, including ResNet158, EfficientNet-b1, -b4, and -b7, DenseNet161 and −201, all pre-trained on ImageNet, were used [13]. RGB intensity values of the H&E images were normalized during training, and data augmentation was performed by performing random rotations (90, 180, and 270 degrees) or horizontal or vertical flips. Binary cross-entropy was used as a loss function, combined with the ADAM optimizer, using a learning rate of 1 × 10^−5^ and weight decay of 0.1 every 10 epochs. Networks were trained for 30 epochs, and training was stopped if the validation error did not improve for 7 epochs.

After training, the test set predictions were made using a sliding window approach, followed by combining neighboring patches to a full ROI prediction. To avoid stitching artifacts in the reconstructed prediction, we generated partly overlapping patches. The weighted average of the segmentation probabilities was calculated and converted to either binary or RGB prediction mask images, essentially as described by Cui et al. [14]. The prediction accuracy of the test set was calculated for each class separately and expressed as an F1 (also known as Dice) score. Machine learning and data handling were performed with Python 3.6 and Pytorch 1.7 using one RTX3090 graphics processing unit with 24 GB of internal memory.

## 3. Results

### 3.1. Dataset

Histopathological H&E-stained slides of 65 pancreatic cancer resection specimens were collected. One specimen was not included because it was not feasible to segment it manually due to its growth pattern. Of the 64 remaining specimens, 43 patients were treated with FOLFIRINOX, 19 with gemcitabine-based chemoradiotherapy, and 2 with gemcitabine in combination with nab-paclitaxel. Table 1 details all the included cases. From the 64 specimens, 50 were used for training, 5 for validation, and 9 as a test set. Of all the generated patches, approximately 3000 contained normal ducts against 6000 with tumor tissue and 5000 with the remaining class, NTET. In total, 14,328 patches were used for training and 2244 for validation.

### 3.2. AI-Based Histopathological Classification of Pancreatic Tissue

The accuracy of the obtained algorithms for segmenting the tumor, normal pancreatic ducts, and NTET in the H&E-stained histopathological sections is summarized in Table 2. The differences between the best-performing models were minor, and the best mean results were obtained when using the U-nets with a DenseNet161 encoder. The best results for tumor segmentation only were obtained using a U-net trained using a ResNet152 encoder. Notably, the modified U-nets with EfficientNet encoders performed relatively poorly; the mean F1 scores for the tumor class and NTET were the lowest of all investigated models. Moreover, these latter models appeared unable to segment normal ducts. The classic U-net performed better than the EfficientNet variants but was approximately 5% less accurate than the DenseNet or ResNet variants. In Figure 1, representative examples of three different cases of the AI-based segmentation results on previously unseen test samples are illustrated. In the figure, the H&E staining, the ground truth (as labeled by the expert pathologists), and the model’s prediction (U-net with DenseNet161 encoder) are illustrated. From this figure, it can be appreciated that the major (and large) structures were recognized well, but clearly, some discrepancies between the ground truth labeling and the network’s prediction were present.

### 3.3. Discrepancies between Ground-Truth and AI-Based Predictions

To gain insight into the errors of the trained DenseNet161 network, the discrepancies between the ground-truth and AI predictions were carefully optically compared with the region of interest in the corresponding H&E section. Several kinds of mismatches were observed. First, the model frequently correctly identified structures that were not delineated during the annotation process, such as tumor buds (Figure 2A–C) or islets of Langerhans (Figure 2D–F). Second, the model incorrectly marked non-cancerous structures as cancerous. Figure 2G–I shows how the model incorrectly predicted ductal metaplasia as malignant ducts. On other occasions, the model erroneously marked vascular structures and folding artifacts as malignant ducts. Finally, the model also classified cancerous structures as non-cancerous, as illustrated in Figure 2J–L.

## 4. Discussion

This study shows that it is feasible to segment the residual tumor burden of PDAC after NAT using an AI-based approach. The highest mean segmentation accuracies were obtained with modified U-nets, especially when trained with a DenseNet161, −201, or ResNet152 encoder, all of which performed better than the standard U-net. Modified U-nets with an EfficientNet-b1, -b4, or -b7 encoder, on the other hand, performed considerably worse when compared with all other tested models, including the standard U-net.

To the best of our knowledge, this is the first report on an AI-based segmentation model to identify residual tumor burden after NAT in PDAC. There are AI-based segmentation models for colorectal, prostate, breast, liver, gastric, squamous, and basal cell cancers published in the peer-reviewed literature [15,16,17,18,19,20,21,22,23,24,25,26,27,28,29,30,31,32,33,34,35,36,37]. Clinical purposes for these segmentation tools vary, including detection and diagnosis, Gleason grading for prostate cancer, and prognostic or predictive feature extraction. In one study, an AI-based segmentation tool was used to measure the response to chemoradiotherapy in a cohort of patients with hepatocellular carcinoma [28]. Moreover, one study reported a segmentation tool that was developed in non-neoadjuvantly treated pancreatic cancer tissue [38]. This model reached a pixel-based precision and recall of 98.6% and 95.1%, respectively. At first sight, these scores appear to be better than our results in the present study. However, these data were obtained on consecutive slides of a single case. Therefore, their model is not directly comparable to ours. Their model was not developed to recognize the wide variety of heterogeneous presentations of PDAC after NAT, but rather to create a three-dimensional representation of histopathological micro-anatomy of one particular case. Therefore, model generalizability is likely to be poor. In terms of performance to similar studies in different cancer types, our model, with a mean F1 score of 0.86, compares favorably against the published performances, ranging from 0.353 to 0.9243 and a median F1 score of 0.835 [15,16,17,18,19,20,21,22,23,24,25,26,27,28,29,30,31,32,33,34,35,36,37]. When comparing our approach to quantifying residual tumor burden to current TRS systems in pancreatic cancer, several take a similar approach [9,10,39,40]. These systems aim to objectively quantify tumor burden and demonstrate favorable performance in predictive ability and reproducibility compared with some of the most-used TRS systems [9,10,40,41]. These studies indicate that objective quantification of residual tumor burden potentially forms a good basis for TRS. Still, for these systems, interobserver variation originating from human assessors remains an issue.

Although the highest unweighted mean (three class) accuracy was seen when we used modified U-nets with a DenseNet161 or −201 encoder, the difference with a ResNet152 encoder was slight. If we only considered the accuracy for identifying tumor tissue in the PDAC samples, the ResNet152 encoder, on average, marginally outperformed the DenseNet variants. When comparing mistakes made by these models, they each exhibited all error types, as shown in Figure 2. However, it should be noted that when we compared these three models (DenseNet161, DenseNet201, and ResNet152), there was not one model that performed best in all individual test cases. In some test cases, the DenseNet variants performed best, while in others, the ResNet variant achieved the highest scores. Given that no one model performed best in all individual test cases, we anticipate that the observed differences in the mean F1 scores were mainly the result of the composition of the relatively small test set, which may be more favorable to a single model. As such, we anticipate that performance differences may diminish when a larger and thus more representative test set is used.

On the other hand, we did consistently see that DenseNet- and ResNet-modified U-net models always performed better than the standard U-net. Interestingly, not all modified U-nets performed better than the standard U-net. All evaluated EfficientNet models performed considerably worse. The reason for this was not apparent. Still, maybe this had to do with the class imbalance in our dataset and that modified U-nets with EfficientNet encoders have more problems with underrepresented classes. Although we attempted to limit class imbalance during the creation of patches for training, it was a given that normal ducts in PDAC samples were relatively rare and not evenly distributed between individual cases.

Across all trained models, we always observed the lowest F1 scores for the normal duct class. This could be partially explained by the previously mentioned class imbalance in our training set. However, it is also important to note that even for the (experienced) human eye, it can be difficult to classify a specific duct as normal or abnormal solely based on morphology. Sometimes, aspects other than cytonuclear atypia are considered, like the architecture, the position within the tissue, and the surroundings. Therefore, it might not be surprising that an algorithm also has difficulty learning subtle characteristics of some malignancies.

The ground-truth annotations must be as accurate as possible to determine model performance. In retrospect, evaluation of the discrepancies between the manually annotated ground-truth and AI-based predictions revealed the manual segmentations to be suboptimal in several respects. In various instances, cancerous tissue was annotated inaccurately or not annotated at all. The latter was, for example, the case with tiny tumor buds, which were often correctly recognized by the algorithms but not manually annotated. In addition, other structures like the islets of Langerhans were occasionally not annotated. The lack of annotation of these structures was probably because annotations of small structures are very labor-intensive and tedious. On the other hand, the dignity of some structures was uncertain and therefore not annotated. It is difficult to predict the effect of these errors on the calculation of the F1 score or even on training the network. Structures missed in the ground-truth annotations but correctly predicted by the algorithms will therefore downgrade the F1 scores. Still, careful identification of the mismatches between the ground truth and the algorithm’s prediction can help to improve the ground-truth annotations of our dataset, which ultimately will lead to improved performance of the algorithm, a technique known as active learning [42].

While the current results are promising, the present study has several limitations, and there is still plenty of room for improvement in segmentation performance and generalizability. For generalizability, it should be noted that the current dataset is relatively small and that all data came from the same laboratory. Additionally, all sections were digitized on only one type of scanner, and the same two pathologists essentially did all the annotation and classification. To develop generally applicable algorithms that accurately identify PDAC in histological samples, more data from different (international) laboratories with patients who received various neoadjuvant regimens must be included. Further improvement can be obtained if the annotation and classification of the pathologic structures are jointly performed by a larger group of expert pancreatic pathologists, and each tissue section is evaluated and classified by multiple experts. Indeed, tissue regions were frequently encountered that could not be classified as either normal or abnormal. Under these circumstances, the study could benefit from a consensus diagnosis from different experts. We should also note that we trained the current models on a relatively small dataset, and all cases were relatively well-differentiated, allowing manual delineation. It was practically impossible to manually annotate tumors with complex and reticular growth patterns accurately, and therefore, one of these cases was not included in the current dataset. Thus, the reported model performance might be optimistic for poorly differentiated tumors. Finally, to better understand factors influencing the prediction accuracies, such as tumor morphology, staining quality, and the age of the archived samples, future studies should ideally provide model performance data on each sample.

In addition to developing more generalizable and better-performing models using larger datasets, future research should optimize the data preparation and AI workflow. The quality of the ground truth could be improved by researching techniques such as antibody-supervised learning [43]. Immunohistochemistry-stained sections can typically provide better contrast compared to H&E-stained sections. This improved contrast can aid in identifying cells that could be mislabeled otherwise and may help to more easily create or automatically generate annotations that follow the contours of structures very precisely. This technique is especially relevant in cases with diffuse cancer growth and an abundance of solitary cancer cells or tumor buds. Moreover, though very labor-intensive, the quality of the ground truth could be further optimized using consensus-based segmentation of the same cases by multiple pathologists. To expand on this in the view of (international) collaborative research, we could see additional benefits in online accessible segmentation platforms. Finally, with the rapid development of AI, researchers should stay vigilant for novel approaches to developing segmentation models. These approaches could involve ensemble learning techniques, weight optimization techniques, alternative data augmentations, and segmentation approaches based on active, semi-supervised, or unsupervised learning techniques.

Ultimately, a new AI-based TRS system quantifying residual tumor burden may address the issues of currently used TRS systems such as subjectivity, handling with response heterogeneity, and the inherent complexity of recognizing and quantifying diffuse cancerous growth and solitary cancer cells. We hypothesize that solving these problems will likely improve the clinical value of TRS. Next to providing a basis for an objective TRS system, we anticipate that this segmentation model could form the basis for tools that can extract relevant tissue features from segmentations on images to gain insight into cancers’ molecular backgrounds. Insight into the molecular backgrounds of individual cancers could aid patient stratification by, for example, identifying novel biomarkers that correlate to the clinical outcome or by identifying molecular tumor subtypes with differential responses to neo(adjuvant) therapies [44,45,46].

## 5. Conclusions

We demonstrated that AI-based segmentation of residual tumor burden in pancreatic cancer after NAT is feasible and may form the basis for an objective TRS system. Further research is required, focusing on the development of extensive training databases and improving training data quality.

## Figures and Tables

**Figure 1 cancers-13-05089-f001:**
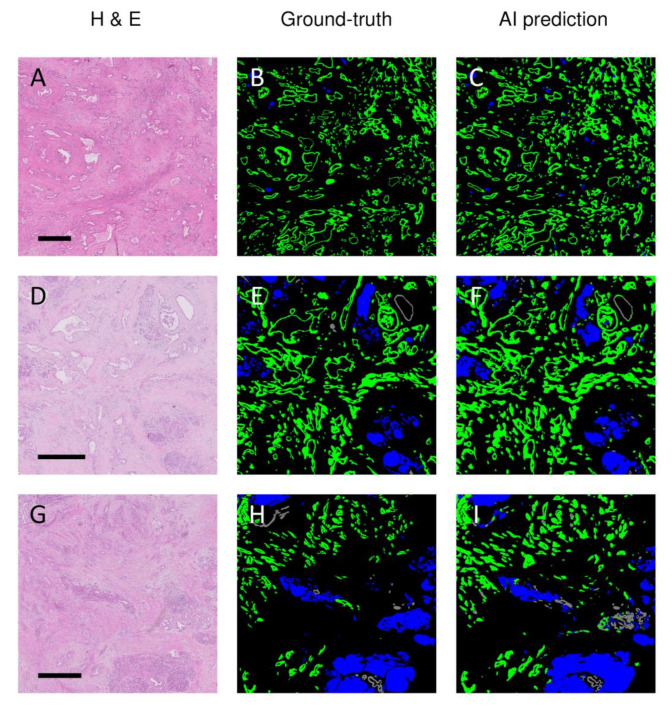
Representative examples of AI-based predictions versus pathologist-based ground-truth annotations of three different (previously unseen) patients with pancreatic ductal adenocarcinoma after neoadjuvant treatment. Legend: H&E = hematoxylin and eosin staining (**A**,**D**,**G**); Ground truth = annotations by the pathologist (**B**,**E**,**H**); AI prediction = prediction based on U-net with DenseNet161 encoder (**C**,**F**,**I**); green = cancerous ducts and in situ neoplasia; gray = normal ducts; blue = remaining non-tumorous epithelial tissue. The scale bar indicates a length of 1000 µm.

**Figure 2 cancers-13-05089-f002:**
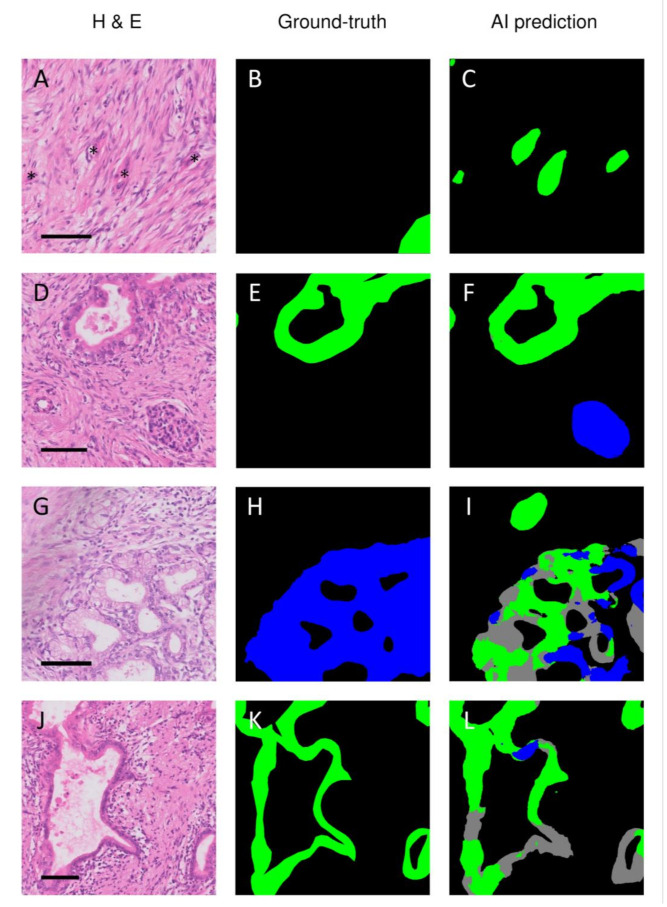
Discrepancies between the ground truth annotations and AI-based prediction of pancreatic ductal adenocarcinoma. Legend: The network frequently correctly recognized structures that were missed during annotating, such as tumor buds (**A**–**C**) or islets of Langerhans (**D**–**F**). The neural network also made classification errors, such as classifying atrophic metaplastic epithelium as either normal or tumorous ducts (**G**–**I**) or classifying tumor ducts as being normal (**J**–**L**). H&E = hematoxylin and eosin; Ground-truth = annotations by the pathologist; AI prediction = prediction by the artificial intelligence model; green = cancerous ducts and in situ neoplasia; gray = normal ducts; blue = remaining non-tumorous epithelial tissue. The asterisks in A indicate cancerous tissue that was missed during the annotation process. The scale bar indicates a length of 100 µm.

**Table 1 cancers-13-05089-t001:** Patient characteristics.

Characteristic	Number (*n*)	Percentage (%)
Tumor Location		
Head	54	84.4
Body	6	9.4
Tail	4	6.3
Neoadjuvant Therapy		
FOLFIRINOX × 8	20	31.3
FOLFIRINOX × 4	20	31.3
Gemcitabine × 3 + RTx × 1	19	29.7
Gem-nab-paclitaxel	2	3.1
FOLFIRINOX × 6	1	1.6
FOLFIRINOX × 2	1	1.6
FOLFIRINOX × 1	1	1.6

Legend: FOLFIRINOX = combined therapy of irinotecan, 5-fluorouracil, leucovorin, and oxaliplatin; RTx = radiotherapy.

**Table 2 cancers-13-05089-t002:** Obtained F1 scores using different U-net encoders.

Encoder	Tumor (F1, 95% CI)	Normal Ducts (F1, 95% CI)	NTET (F1, 95% CI)	Mean (F1)
DenseNet161	0.86 ± 0.09	0.74 ± 0.12	0.85 ± 0.07	0.82
DenseNet201	0.85 ± 0.09	0.77 ± 0.13	0.85 ± 0.08	0.82
EffecientNet-b1	0.78 ± 0.15	0	0.77 ± 0.13	0.51
EffecientNet-b4	0.77 ± 0.14	0	0.61 ± 0.73	0.46
EffecientNet-b7	0.81 ± 0.12	0	0.82 ± 0.12	0.54
ResNet152	0.88 ± 0.06	0.77 ± 0.14	0.73 ± 0.15	0.79
None (‘standard’ U-net)	0.83 ± 0.10	0.69 ± 0.23	0.83 ± 0.15	0.78

Legend: F1 = Dice score; 95% CI = 95% confidence interval; NTET = remaining non-tumor epithelium; Mean = unweighted average of F1 scores for the tumor, normal ducts, and NTET.

## Data Availability

The data used to support the findings of this study are available from the corresponding author upon reasonable request.

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
