# Peer review of "Artificial Intelligence-Based Segmentation of Residual Tumor in Histopathology of Pancreatic Cancer after Neoadjuvant Treatment"

_cancers, 2021, doi:10.3390/cancers13205089_

Round 1

Reviewer 1 Report

The authors present a proof-of-concept for the development of an AI-based segmentation protocol for histopathology of pancreatic ductal carcinoma, that could function as a more objective TRS system. For this they select 64 archived samples from which they digitized one H&E stained slide per sample and compared segmentation predictions by several modified U-net models to a ground-truth set by pathologists. 
Their findings show that, at least for this cohort, several models are able to predict segmentation that compares relatively well to the ground-truth.

All in all this is an interesting approach with potentially valuable clinical outcome. There are however several limitations to this study, a lot of which the authors already mention in the discussion. 

Minor comments

lines 38-39: the abstract describes best results with a DenseNet166 encoder, while line 115 in the Materials does not include this encoder and the results in line 144 mention DenseNet161 obtained best results. The DenseNet166 encoder then re-emerges in line 188 of the discussion.

line 119: please use another representation of 1 x 10-5 (with a higher placed dot for example)
line 153: 'Unet', please stay consistent with the 'U-nets' in the rest of the text

line 157: table 2, some decimals are separated by '.' other by ','

line 215: 'Densenet201' stay consistent with 'DenseNet201' in the rest of the text.

With these previous remarks in mind I suggest the authors read through the manuscript to increase uniformity of the text.

line 81: could the authors comment on whether the selected samples (64) were the only neoadjuvantly treated pancreatic cancer specimens available or if they were selected from a larger archive. 
If the latter, what were their criteria for selection (I believe lines 266-270 hint towards this). 

lines 214-217: regarding the comparison of DenseNet161, Densenet201 and, ResNet152. 
Are the mistakes made by the models similar or different in nature. 
By this I mean: Figure 2 shows discrepancies between DenseNet161 predictions and ground-truth that effect F1. 
Do the other models exhibit similar discrepancies (and their frequencies) or is model-bias in discrepancies observed? Also, have the authors considered stacked modeling, to improve their predictions? Or optimization of the weights before averaging segmentation probabilities?

lines 217-219: 'Observed differences in mean F1 scores are thus 217 mainly the result of the composition of the test set, and not primarily due to the perfor- 218 mance of one of these models.'
Do the authors hereby mean that different U-nets were trained with different training sets (and thus perfomances was evaluated on a different test set)? If so this should be mentioned in the Methods. 
Can the authors comment on the relative performance of these models when using the same test, validation and training sets for all models?

Given the fact that DenseNet161, Densenet201 and, ResNet152 perform very similar in terms of F1, can the authors comment on their computational needs (CPU time, memory needs, etc)?

It would be interesting to see a table in supplementaries describing each models performance on each individual sample annotated with more details on the sample itself, such as morphology or quality of the staining, age of the archived sample, number of patches in each class, etc. 
This in order to better understand factors influencing prediction accuracy.

Can the authors comment on the hyperparameter selection (epochs, optimizer, learning rate, etc)

Have the authors considered random cropping considered for data augmentation?

Reviewer 2 Report

Janssen et al in this manuscript present their results on first insights into the use of AI-based algorithms for assessment of tumor response in clinical PDAC samples. With the emergence of neoadjuvant/induction chemotherapy response scoring is a relevant clinical topic in treatment of PDAC. In this regard, their work mostly falls into the category of a “proof of principle” – this, of course, comes with some limitations, which the authors correctly point out in the discussion. Generally, I consider the manuscript well worthy to be published in Cancers, however, I have some questions and suggestions for minor revisions:

  • As the authors point out themselves, interobserver reproducibility for some TRS systems has been shown to be poor – might AI-based methods be able to overcome this so regression grading, instead of quantification of residual tumor burden, might actually be back in the game again at some point? Especially, as is mentioned in the discussion, based on antibody applications?
  • In the same line, for the non-pathologists I would suggest elucidating the classifications used for the annotations in round one and two by correlating them roughly to the current tumor response scoring systems. I would like to understand better which items of the respective TRS would show up in which class of annotations. The connection between classifications for annotations and the endpoint of residual tumor burden is never made in the article. Aren’t there any data from this study on at least the range of residual tumor burden in the slides?
  • I would be interested on a comment on intratumoral heterogeneity – how many sites within the tumor would need to be sampled to give a representative result? Similarly, different chemo regimens can affect different compartments differently - would this pose a problem for machine learning?
  • The article mentions in the discussion that poorly differentiated tumors were excluded – this should be contained in the methods section
  • The authors have done important work in the field before which is rightfully cited by themselves. Nevertheless, as not only a new technique of assessment but furthermore a new system of TRS is used, I believe for the information of the readers adding other groups’ work would be interesting – e.g. Okubo, S. et al., Sci. Rep. 9, 17145 or Chou, A. et. al., Am. J. Surg. Pathol., 25.3 (2021): 394 – 404
  • Finally, language-wise there are few sentences I stumbled over, please review:
    • „remaining non-tumorous epithelial tissue 104 (NTET) consisting of islets of Langerhans, acinic tissue, and atrophic mesenchymal parenchyma“ metaplastic parenchyma?
    • “Even if present in a sample, only a minor population of the epithelial tissue could be attributed to the normal duct class” – do you mean to state normal ducts were rare, if any were found at all? I find the sentence difficult to understand.
    • “future researchers should stay vigilant” – probably not only future but also current researchers?
    • “aid in identifying cells that you may have mislabeled otherwise” style is not consistent
    • “Insight into a cancers' molecular background“
